# A Research Roadmap for Sustainable Design Methods and Tools

**Jeremy Faludi [1,\*], Steven Hoffenson [2], Sze Yin Kwok [3], Michael Saidani [4], Sophie I. Hallstedt [3], Cassandra Telenko [5] and Victor Martinez [6]**

[1]  Faculty of Industrial Design Engineering, TU Delft, 2629 HS Delft, The Netherlands
[2]  School of Systems & Enterprises, Stevens Institute of Technology, Hoboken, NJ 07030, USA; shoffens@stevens.edu
[3]  Department of Strategic Sustainable Development, Blekinge Institute of Technology, SE-371 79 Karlshamn, Sweden; sze.yin.kwok@bth.se (S.Y.K.); sophie.hallstedt@bth.se (S.I.H.)
[4]  Industrial Engineering, Université Paris-Saclay, CentraleSupélec, F-91192 Paris, France; saidani.michael@gmail.com
[5]  Ford Motor Company, Detroit, MI 30170, USA; ctelenko@ford.com
[6]  Wilson School of Design, Kwantlen Polytechnic University, Surrey, BC V3W 2M8, Canada; victor.martinez@kpu.ca
\*  Correspondence: j.faludi@tudelft.nl

**Abstract:** Sustainable design methods and tools abound, but their implementation in practice remains marginal. This article brings together results from previous literature reviews and analyses of sustainable design methods and tools, as well as input from design researchers and professional practitioners to identify the needs and gaps in the area. It results in a shared vision of how sustainable design methods and tools can be more tightly integrated into mainstream product design and development, as well as the current state of practice and research in relation to four central questions: What are the needs and values of industry regarding sustainable design? What improvements in sustainable design methods and tools would most drive industry forward? How should researchers move forward with developing more useful sustainable design methods and tools? How can sustainable design be more effectively integrated into industry? A roadmap for the international sustainable design research community is proposed with descriptions of short-, medium-, and long-term tasks for addressing each question. The purpose is to support collective progress and discussions on method and tool development and adoption, and to enable more tangible success in mainstreaming sustainable design practices in industry.

**Keywords:** sustainable design; design methodologies; product development; design methods and tools; research agenda; industry adoption

## 1. Introduction

### 1.1. Context and Motivation

Sustainable design (SD), also known as design for sustainability or sustainable product development, aims to transform product development practices to enable all species to flourish for all time. This is often operationalized as optimizing the environmental and social wellbeing that results from the life cycle impacts of products, systems, and activities [1–3]. Sustainable production, use, and end-of-life begins with sustainable design, considering its three pillars, namely environmental (e.g., through eco-design), economic (e.g., through design for green profit), and social (e.g., through design for social sustainability). The United Nations' Sustainable Development Goals

(SDGs) [4] provide one framework, with targets and metrics for the year 2030. Particularly relevant are SDG 12 and SDG 9. SDG 12 is "Responsible consumption and production," including the following targets: "by 2030, achieve the sustainable management and efficient use of natural resources," "encourage companies to adopt sustainable practices and to integrate sustainability information into their reporting cycle," and "develop and implement tools to monitor sustainable development impacts." SDG 9 is "Industry, innovation and infrastructure," including the targets: "promote inclusive and sustainable industrialization" and "by 2030, upgrade infrastructure and retrofit industries to make them sustainable, with increased resource-use efficiency and greater adoption of clean and environmentally sound technologies and industrial processes." These goals were intended to drive government legislation and company policies, but even if such legislation were enacted, how could these goals be achieved?

Academics, activists, and proactive companies have suggested SD practices for fifty years or more [5,6]. Since the 1990s, practitioners in academic, corporate, governmental and non-governmental roles have developed a broad but disjointed collection of SD practices with different motivations, scopes, and applicability. Recent surveys identified over 600 unique eco-design tools or methods [7,8]. These eco-design, circular or sustainable design, and other "design for X" approaches have been developed and extensively reviewed by scholars [9–18]. They include a broad variety of practices, including multi-step methods, software tools, simple activities, broad mindsets, and checklists of goals; for inclusivity, this project refers to all of these practices as "sustainable design methods and tools" (SDMTs). Among these SDMTs, no clear, ubiquitously adopted SD practices have emerged. Rather, every designer or team seems to take a unique approach to design that aligns with their work structure, values, domains, and expertise. In addition, different companies and work cultures make their own tools while scholars and graduate students develop other new methods and tools, often without company contact or testing. This leads to redundancies, wasted time, and likely sub-optimal practices. Despite the proliferation of SDMTs through a wide variety of forms (guidelines, checklists, software, cards, mindsets, etc.), their uptake by industry has been relatively low [11,18–20].

With this lack of uptake in mind, the Design Society's Sustainable Design Special Interest Group (SD SIG) held a workshop at the 2017 International Conference in Engineering Design (ICED) to discuss SD challenges, opportunities and directions. This meeting of 40-50 participants from the design research community concluded with several broad goals for the future of research in SDMTs, including the following: create a consolidated database of existing tools for supporting future research, reach consensus on which tools and models to ground future research, publish tools and case-studies in open-access or popular outlets, and define and improve user-friendliness and ease-of-adoption of tools. A working group of international scholars formed afterward, with the mission to build on existing efforts and guide the future of SDMT development.

*1.2. Objectives*

This article proposes a new research roadmap to provide vision and guidance for SD research, to help drive sustainable design into ubiquitous use and create global-scale change in how products are created, used, and managed at all life-cycle stages. Its goals are to increase the impact of SD research by driving company adoption and integration of sustainability into standard product development processes. This work brings together previous reviews and analyses of SDMTs to identify areas of consensus and suggest future research and development directions. The roadmap is intended to guide the next decade of SD research by galvanizing the international research community toward common goals of developing and integrating SD principles, methods, and tools into mainstream product development.

The roadmap's primary audience is academics researching design in industry practice. Government and non-governmental organizations (NGOs) sometimes take roles similar to industry or academia, so the roadmap may also contribute to their work and vice-versa. They create guidelines, tools, and regulate systems. This roadmap uses the terms "industry practice" and "academic research" because they are the most prevalently researched to date, and because industry

is the main producer of physical products and many services. Writing a roadmap for government, NGOs, and other audiences should be done by experts in those fields, but we hope these findings may be useful to anyone who influences the design of products and systems to lead society toward a healthy, just, and abundant world for all.

This research roadmap takes the form of key research questions for design-related academics and others to pursue, broken down into more specific subtopics and into short-, medium-, and long-term tasks. Each task may be a Ph.D. thesis, sponsored project, or other collaboration between universities and companies to ensure the design and development of sustainable products and services. This article's structure focuses more on the roadmap than an extensive literature review. The following sections describe the methodology in detail, then results are subdivided into the vision, baseline situation (listing most literature), and the roadmap itself; a discussion follows, including limitations, and lastly the conclusion suggests next steps.

## 2. Methodology

To establish a structured, evidence-based roadmap for SDMTs, several steps were taken to understand the state of the art, gather expert feedback, generate ideas, and refine a path forward. This paper reports findings from a combined descriptive and prescriptive study, inspired by the Design Research Methodology [21]. The roadmap was formulated using the "backcasting" methodology from the Framework for Strategic Sustainable Development [22] or The Natural Step [23], which starts with a vision, compares it to the current reality (the baseline), ideates solutions, and chooses a path forward (the roadmap). This process was also informed by roadmap development guidelines of Simonse [24] and especially Kim's guidelines for developing roadmaps in a "volatile, uncertain, complex, and ambiguous" world [25]. Figure 1 provides an overview of the major phases of this process, which includes five high-level steps: problem definition, needs and gaps identification, roadmap scoping, roadmap definition, and expert feedback.

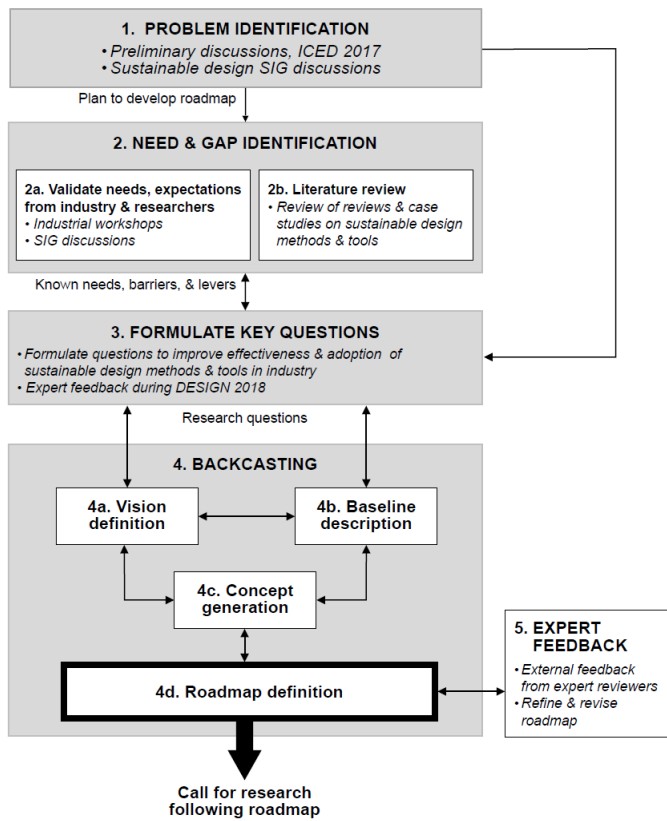

**Figure 1.** Overview of the methodology.

This research was conducted by an international working group formed at the ICED 2017 workshop. The roadmap developed over two years through the combination of 39 one-hour working group meetings to share, critically evaluate, synthesize and curate the results of 2–6 weeks of individual work between meetings. The working group consisted of 8 academic and industrial researchers who hold PhDs and practice in the field of sustainable design. The team brought unique insights and diverse perspectives, including:

- Global work experiences in Canada, Denmark, France, Hong Kong, the Netherlands, Sweden, the United Kingdom, Singapore, and the USA
- Complementary positions including assistant and associate professors, industry researchers, and postdoctoral researchers
- Diverse areas of expertise including circular economy, eco-design, design for sustainable behaviour, design optimization, environmental assessment, environmental management, sustainability certifications, sustainable product development, system modeling, mechanical engineering, and user-centered design.

## 2.1. Step 1: Problem Definition

The first task was to identify and establish the overarching goals of the sustainable design research community. During ICED 2017, the SD SIG convened a workshop of 40-50 design researchers and professionals to brainstorm and synthesize a list of challenges and goals for the community, summarized in Table 1. This collective description of the goals for the sustainable design community was synthesized and later expanded by the working group into a vision of an ideal future, which describes a world where sustainable design is a widespread practice in industry (Figure 1(4a), detailed in Section 3). Similarly, the collective description of the current sustainable design situation led to the description of the baseline situation (Figure 1(4b), detailed in Section 4).

**Table 1.** Collective description of the state of sustainable design from ICED 2017 SD SIG workshop.

| Situation | Goals |
|---|---|
| • There are many SDMTs, most of which have low adoption rates in industrial or business practice.<br>• Practitioners do not have access to or awareness of all of the academic tools.<br>• Different companies or cultures might prefer adopting different tools or developing their own in-house tools.<br>• There is a general lack of resources for practical SDMT implementation (e.g., human resources, tools to invest in, access to experts).<br>• Case studies have historically enabled adoption.<br>• Many challenges inhibit companies from embracing sustainability, including how to quantify, what to measure, and how to communicate.<br>• Companies have difficulties translating SD to monetary business needs. | • The SD community should create a 10-year vision, with consensus goals that direct work and avoid redundancies.<br>• Companies should be able to easily adopt, modify and combine tools.<br>• SD practices must take into account and accommodate needs for diversity.<br>• The SD SIG should increase education and training about sustainable design.<br>• Industry should have easy access to the tools and understand the criteria for when and where to use them.<br>• Tools should be easier to apply and compatible with existing business and design methods and processes.<br>• The SD SIG should increase the uptake and appropriate use of SDMTs in industry.<br>• SDMTs should be developed and framed in ways that contribute to the UN SDGs. |

## 2.2. Step 2: Need and Gap Identification

The second step in the process (Figure 1(2)) was to identify the needs and gaps related to SDMT development and use as they relate to both industry practice and academic research (Figure 1(2a)). This corresponded to step one of Kim's roadmapping process: data gathering [25]. The working group reviewed the literature individually, compiling papers under categories of review papers and

case studies in shared folders and into an annotated bibliography spreadsheet. The findings were discussed as a group to synthesize and expand (Figure 1(2b)). Initial papers reviewed knowledge and methods in sustainable design, but expanded as each member independently responded and linked papers in the spreadsheet to the research questions through an iterative process from Step 3 (Figure 1). Each member of the working group contributed review papers on SDMTs with which they were familiar, approximately 25 unique papers, and additional articles were found through searches in Web of Science, Science Direct, and Google Scholar, as well as papers that either cited or were cited by those that were already part of the review, resulting in over 100 papers. Many search terms were used, including "sustainable design methods," "green design guides," "eco-design tools," "sustainable product development practices," "integration of sustainability into design strategy," and more, including variations thereof. Literature was selected if it related to improving sustainable development of products and services, or barriers to that development; literature on government policy, environmental science, finance, marketing, supply chain management, and other disciplines were not selected unless they also related directly to the practice and management of design. From this review of reviews, key knowledge gains and gaps emerged, as detailed in the Baseline (Section 4).

### 2.3. Step 3: Key Questions Formulation

The third step (Figure 1(3)) synthesized needs of researchers, industry, the planet, and society into key research questions to pursue. This corresponded to step two of Kim's roadmapping process, extracting core design principles. While substantial advances in sustainable design practices have been made in recent decades, open questions remain about how to implement large-scale improvements across the diverse spectrum of industries. Starting with the list of goals from the ICED 2017 workshop (Table 1), each group member used their individual expertise, the literature review and discussions, to generate research questions. Combined, this individual brainstorm resulted in 17 key questions, presented in Appendix A. These 17 questions were synthesized and reduced to more essential questions through discussions with experts at the DESIGN 2018 conference, and five subsequent working group meetings to yield six more focused questions: three about the nature of industry practice, and three about the state of research practice, presented in Appendix B. Finally, during the roadmap definition process (Step 4, below), these were refined into four interrelated research questions to structure the roadmap:

**RQ1:** What are the needs and values of industry regarding sustainable design?
**RQ2:** What improvements in SDMTs would most drive industry forward?
**RQ3:** How should researchers move forward with developing useful SDMTs?
**RQ4:** How can sustainable design be more effectively integrated into industry?

### 2.4. Step 4: Roadmap Definition Using a Backcasting Approach

Next, the roadmap was designed using backcasting (Figure 1(4)), inspired by the "ABCD" method in the Framework for Strategic Sustainable Development [22]. This approach takes four steps: (A) Aspirational vision for success in the system; (B) Baseline—describe current conditions and the gaps between them and the aspirational vision; (C) Creative solutions to bridge the gaps; and (D) Decide on which creative solutions to use and connect as paths forward from the baseline to the vision. The final set of paths are the roadmap. Each working group member was assigned two of the six research questions and tasked with applying the ABCD approach. Each question had two members assigned to collaborate, writing a two-page evidence-based argument, with a third member to review in detail. The results were presented, evaluated, and discussed in working meetings.

Backcasting was preferred to other roadmapping processes because of the priority placed on a specific outcome, the sustainability vision. It was similar to Kim's roadmapping process steps two through five, as they identified common themes, narrowed focus, prioritized features that support core values of the proposals, considered how solutions would be applied to meet needs, and laid out roadmap goals in timelines according to their dependencies.

*2.5. Step 5: Expert Feedback*

The final step (Figure 1(5)) was to elicit external feedback from expert reviewers on the complete draft of the roadmap article. Feedback was received on the roadmap from six professors and researchers in sustainable design with experience as practitioners and researchers in Canada, the Netherlands, and Sweden. The working group incorporated this feedback to refine the roadmap.

## 3. Vision: Where We Want to Be

The first step of the ABCD backcasting method is to envision an ideal future with high ambitions, rather than safe achievable goals. Our aspirational vision is that all companies have a long-term sustainability strategy that is deeply integrated into their product development and other business practices so that their products and services enhance the environmental and social health of the world more than they damage it. They work in line with the SDGs, in particular SDG 12 and SDG 9 that champion responsible consumption and production along with sustainable industrialization. Corporate sustainability strategies are enabled and accelerated through well-established and widely practiced SDMTs, and these practices lead to real systemic improvements in the SDG targets and metrics. Researchers and developers of SDMTs work closely with companies for a deep understanding of their needs, and they continuously improve SDMTs to solve evolving challenges and deliver both sustainability and business value. This does not replace independent practitioners creating or modifying SDMTs informally and improvisationally, but heightens the quality of SDMTs by professionalizing the field. Professional designers, engineers, and managers are thoroughly versed in sustainable design practices, as design engineering degrees have deeply embedded sustainability in their curricula; both students and professionals advance their use of and proficiency in these tools through continuing education.

New and updated support tools are easily discovered, built upon, learned, and chosen for relevant use by industry. There exists a consistent and up-to-date means of learning about and contributing to the body of existing SDMTs; one concrete example could be a public database of SDMTs. SDMTs are distinguished based on characteristics such as their objectives, values for sustainability, business benefits such as innovation or cost-cutting, related methods, as well as when and where the SDMT is appropriately applied. SDMTs are tested and rigorously documented by empirical studies of business practice. SDMTs can easily be updated, combined, and supplemented.

The results from the latest developed or improved SDMTs are regularly shared across industry and provide both sustainability and business advantages. In many cases, SDMTs are accompanied by illustrative case studies that help companies understand the benefits and update their sustainability practices. The SDMTs guide companies in developing sustainable innovations, even restorative innovations, and they also support the creation of strategic roadmaps for sustainable development solutions that integrate perspectives from all departments within companies. There is a clear incentive for management and companies to collaborate frequently with researchers on SDMTs, specifically using action research and case studies. Figure 2 presents a summary systemic diagram of this vision and how the research questions fit into the elements and connections in the vision.

Ideally, this vision will be realized by 2030, aligned with the timeline of the SDGs. We recognize that this is ambitious for many of the tasks and objectives described in this article, but swift action is necessary for the health of the planet and society, and there are no inherent technical barriers, only market and institutional inertia. New laws and other external pressures are likely required to motivate industry, though those are outside the scope of this roadmap. Similarly, it is unlikely that all companies will meet these conditions, but the vision is intentionally ambitious, representing an ideal future state. Incomplete achievement of a challenging goal often drives more progress than success at an easy goal.

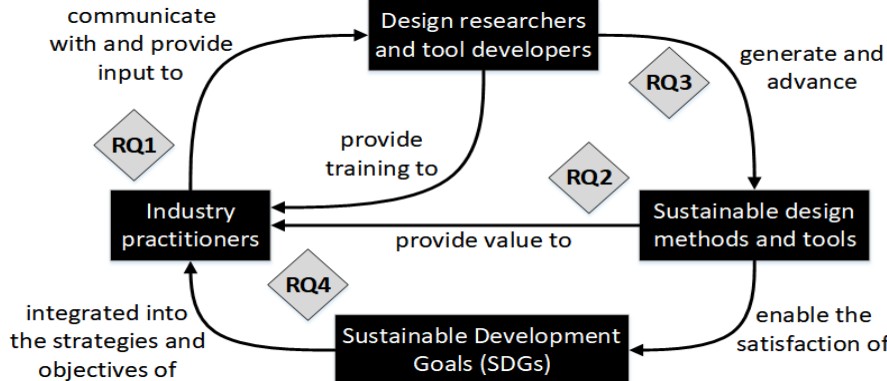

**Figure 2.** Systemic diagram of future vision, with research questions (RQs) mapped among the nodes and links.

## 4. Baseline: Where We are Today

To achieve the vision, the sustainable design community must build from the current state of practice and research. The current reality is not completely antithetical to the vision put forward in Figure 2, but it requires dramatic improvements. The four RQs in this article represent the main barriers that separate the current reality from the future vision. This section presents where we as a sustainable design community are today with respect to the four RQs—referred to as the baseline situation—setting the stage for the research roadmap in the following section.

### 4.1. RQ1: What are the Needs and Values of Industry Regarding Sustainable Design?

Although the research community has formulated a wide range of SDMTs, checklists and guidelines are the most often used in industry, if sustainable design is practiced at all [26]. Regulatory guidelines or guidelines from non-governmental organizations often drive SD action or inaction, due to a mix of monetary or reputation incentives and clear goals/codification [27–29]. Life cycle thinking methods are sometimes employed in the early stages of design, but life cycle assessment (LCA) methods are mostly relegated to material changes and late-stage design, when the design freedom is already rather constrained [11,30]. Rossi et al. [13] reviewed eight categories of SDMTs and identified cost, lack of knowledge, and over-formalization of methods as key obstacles that limit implementation and effectiveness in industry. Lindahl [31] interviewed 12 designers about their experiences and preferences with SDMTs and concluded that tools are generally used when they are simple to use and satisfy the requirements of the organization. Faludi and Agogino [26] interviewed 27 designers, engineers, and managers, finding sustainable design methods are almost never practiced in their entirety, though a few are perceived as useful for innovation in addition to sustainability. Van Hemel and Cramer [32] surveyed 77 small and medium enterprises to understand their use and perceptions of 33 eco-design strategies; they found that non-environmental or customer-demanded benefits guided strategy selection.

Knight and Jenkins [33] assessed the applicability of various eco-design tools and identified that many tools are not adopted due to the need for process-specific customization. According to Wallace [34], in most cases, there is a missing link to concretely transfer design methods into practice. To help with this transfer problem, Zhang et al. [19] developed a navigation framework to align operations, tactics and strategies regarding sustainability integration, finding a gap between corporate performance and product-level sustainable design coordination. Pigosso et al. [12] demonstrated that suitability of different SDMTs depends on competency, or "ecodesign maturity", of project management. Ahmad et al. [16] also state that the maturity of SDMTs must be improved, including being easy to use, resource and time efficient, and able to provide guidance to make sustainability improvements. Faludi [35] tested three sustainable design methods with over 500 professionals and students to find that they valued parts, but not all, of each method, suggesting great room for

improvement and cross-fertilization of SDMTs. Kwok and Hallstedt [36] found that inadequate communication between design teams and consumers cause product developers to inadequately understand consumer needs, and cause consumer confusion about product sustainability and the complexity thereof [37].

*4.2. RQ2: What Improvements in SDMTs Would Most Drive Industry Forward?*

Some of the common themes from the literature and discussions were that SDMTs often fall short in ease of implementation, provision of specific actionable recommendations, rigor of methods and results, transparency of methods, alignment with business incentives, and accompaniment of suitable training in universities and companies to foster industry adoption (see Section 4.4). In analysis of successful environmental regulation, Roxas and Coetzer [29] found that successful sustainability action requires an alignment of a firm's social norms, opportunity to innovate, codification (often provided by regulation), and cognitive beliefs. Rossi et al. [13] identified a need to resolve "the over-formalization of methods and tools in comparison with the complexity of the product development processes, and the consequent divergence between the academic method and the real industrial and designer's need." Robèrt et al. [10] studied the application of SDMTs and their interrelationships and found that companies often have objectives that are too vague for sustainability efforts, and they apply tools detached from a systems perspective or comprehensive strategic planning. Several studies found that most SDMTs are too time-consuming or cost-intensive to learn or perform [38–40], or that they do not fit with existing company practices [32,33].

Consequently, tools with a combination of practical value and a low implementation cost are the most successful. Sustainability is often treated as a bonus feature, thus SDMTs are often only used when time and resources allow. When sustainability practices cut costs significantly, they are often reclassified as value engineering, not sustainability, such as Lean manufacturing [41], or general strategy and innovation [27]; this inadvertently classifies sustainability only as practices that are unprofitable. Certain practices have been identified as "low hanging fruit", and have seen increasing implementation, such as Lean manufacturing, the use of recyclable materials, or meeting eco-label standards [42,43]; these are likely popular because of a combination of their clarity of environmental solutions and their proven economic business cases.

To be effective, SDMTs must change to better balance incentives (such as regulation or business cases), ease of implementation, substantive sustainability improvements, and specificity to the companies' design challenges. Easy-to-implement SDMTs are often based on generalized rules of thumb rather than precise data, making their guidance questionable [44]. These easier SDMTs still suffer from a lack of clear actionable recommendations: general principles such as "eliminate toxicity" can be difficult to apply when qualifying options are rare or unknown, and complex tradeoffs must often be made between energy use, resource consumption, toxicity, and social impacts [45]. This makes simple vague guidelines harder to implement despite their simplicity, and can hamper substantive sustainability improvements. Ease of implementation is further reduced when SDMTs are not aligned with business incentives, i.e., sustainability measures can add cost or impair other factors [39]. Even if companies do allocate resources for sustainable design, selecting and organizing SDMTs to support corporate strategy is still challenging, due to complex contexts [46]. Practitioners' lack of access to the relevant literature or SDMT training can also hamper adoption, as can the complexity of collaboration among stakeholders during product development [19,38,47,48]. To be integrated into product development, SDMTs must also accommodate a company's organization, internal processes, and roles [49].

*4.3. RQ3: How Should Researchers Move Forward with Developing Useful SDMTs?*

Despite the fact that hundreds of SDMTs have been presented in the design and management literature, there is a lack of connection and long-term follow-through between researchers and industry to provide consensus on how to choose the right tool for a project and its context. Without such criteria, researchers cannot consistently evaluate and judge the suitability of tools that they develop. As a result, most SDMTs are created in a vacuum, disconnected from industry realities. In

addition, there is duplication of development effort in addition to gaps of unmet needs. For example, as most tools focus on environmental indicators, there are few social sustainability tools or methods [50]. There are no widely accepted best practices for creating effective SDMTs, nor is there a common resource to help industry choose among existing SDMTs.

To help choose the right tool for a project, several studies have reviewed the available eco-design tools and proposed different taxonomies for characterizing and choosing from among them [8,11,14,15,17,46,51,52]. Some of the main criteria for differentiating among these tools include, for example, difficulty level and time required [11], product development process phases and support processes [12,15], economic activities, and departments in companies [8]. By organizing these eco-design tools, many of these studies draw conclusions regarding the ways that different tools can and should be used as well as the needs for future eco-design tool development [10,15,53]. Some have also empirically tested design methods to obtain feedback on what practitioners do and do not value in them [15,51,54,55] (see RQ1 above), but so far this research has been largely disconnected from the process of developing new design tools or methods, or recommending the right tool for a project.

### 4.4. RQ4: How CAN Sustainable Design be More Effectively Integrated into Industry?

To effectively integrate SDMTs in a company, the tools must be industry-ready as in RQ2 and the company must be inclined to adopt them. It may be the case that existing methods and tools are sufficient already, but they are insufficiently publicized, adopted, required by regulations, or otherwise integrated into industry's cognitive beliefs and social norms of practice. This hypothesis is evinced by studies continuously improving managerial competency and matching SDMTs accordingly [12], and accounting for cognitive and social beliefs in firms when creating regulations and guidelines [29].

Currently, in the authors' experiences, most education for sustainability comes from certification programs or research programs at universities whose graduates are hired to work in consultancies or in dedicated internal sustainability roles. Many of these roles are in departments related to marketing or business strategy and analytics instead of technology and design. The lack of integration of sustainability into engineering and design professions then reinforces the lack of integration of sustainability into standard engineering and design education [56]. There are some media outlets and organizations sharing news and methods for these sustainability practitioners, such as GreenBiz, the Ellen MacArthur Foundation, or Sustainable Brands. There are also some industry focused groups, such as GreenBlue, whose Sustainable Packaging Coalition markets the "How to Recycle" label and engages in U.S. recycling research. Industry associations, such as the International Electronics Manufacturing Initiative, are also actively involved in research and tool development. Achieving widespread adoption requires an education, outreach, and implementation plan. From an education standpoint, it is important to create a pipeline of engineering graduates who are familiar with sustainable design and able to implement best practices and the latest methods and tools. Current industry practitioners must also be made aware of and trained in these techniques, as today's trained graduates can take ten years or more to rise to positions of adequate authority in companies to make change.

To address this issue, actionable and easy-to-adopt processes are needed to facilitate uptake by industry. In recent years, researchers have started to investigate organizational barriers and incentives for sustainability implementation from the perspectives of risk assessment [57,58], portfolio management [59,60], product requirements [61], sustainable policy, regulation [62], and overall company vision and mindset [63]. Further investigations are still needed to deepen our understanding of sustainability implementation in organizations or companies [64].

## 5. Roadmap: How to Achieve the Vision

To move the design research community from this baseline toward the desired vision, a roadmap is suggested, setting a course in four parallel and intersecting paths associated with the four research questions (RQs). Because the goal is for industry product developers to adopt these SDMTs

within design contexts at scale, an analogy is drawn between the four research questions and the Design Thinking, or Human Centered Design, process [65], shown in Figure 3.

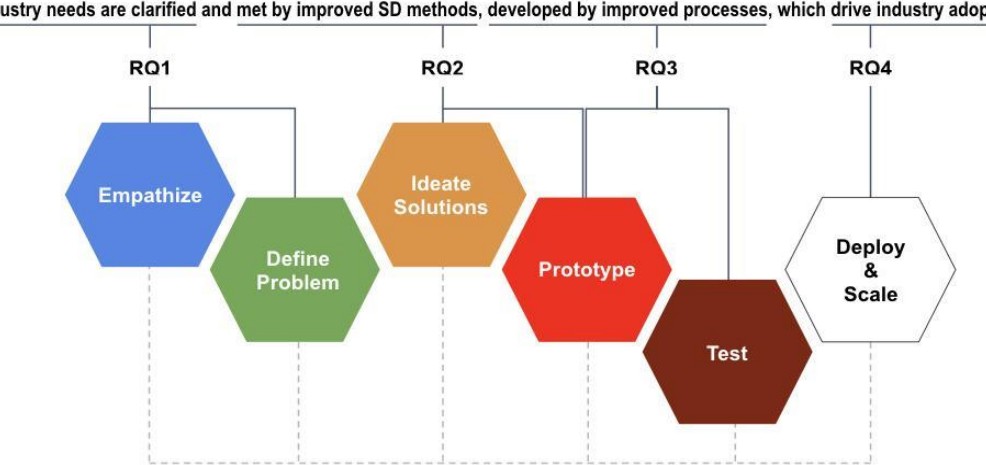

**Figure 3.** The vision and research questions mapped onto the Human Centered Design / Design Thinking process; graphic based on d.school [65].

These four research paths are shown in Figure 4, with each path further divided into themes. Each theme is addressed through short-term, medium-term, and long-term tasks in Figures 5–8; detailed explanatory text follows each figure. There is no fixed timeline associated with these tasks, because they could be implemented iteratively in multiple cycles rather than linearly, and ambitious research groups could push progress in one area faster than other areas. Each theme might be a Ph.D., or each task might be a master thesis, depending on depth. However, to accomplish all the tasks by 2030 would suggest achieving the short-term tasks in 2–4 years, the medium-term tasks in 5–7 years, and the long-term tasks in 7–10 years. Because the goals are ambitious, real timelines may be longer, but due to the urgency of many sustainability challenges, faster progress is preferred.

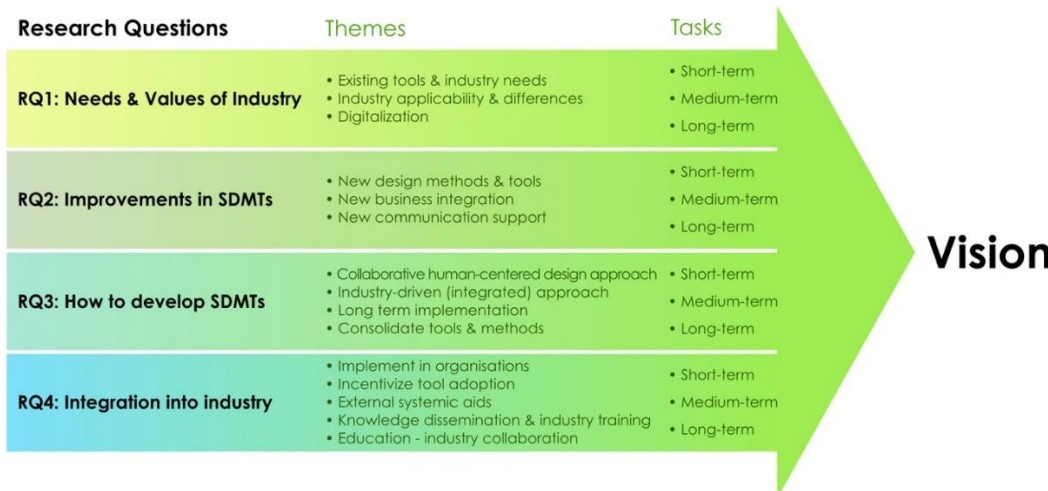

**Figure 4.** Summary of roadmap, listing research questions and themes of the tasks for research and development of sustainable design methods and tools. For lists of tasks, see Figures 5–8.

## 5.1. RQ1: What are the Needs and Values of Industry Regarding Sustainable Design?

The first step toward achieving the future vision of sustainable design outlined in Section 3 is to clarify the needs and values of industry. This will help ensure that future development efforts of SDMTs focus on understanding the roles of the users and decision-makers, and meet their needs without compromising sustainability imperatives. From the literature discussed in the baseline

(Section 4), it is clear that deficiencies of SDMT coordination and access are hindering industry understanding, communication, and adoption. Three main themes are proposed for this RQ: (i) making sense of existing tools and knowledge about industry needs; (ii) increasing industry applicability of SDMTs and adapting to industry differences; and (iii) digitalizing sustainable design resources and making the most of digitalization. These themes and the research guidance proposed for these themes are summarized in Figure 5 and further described below.

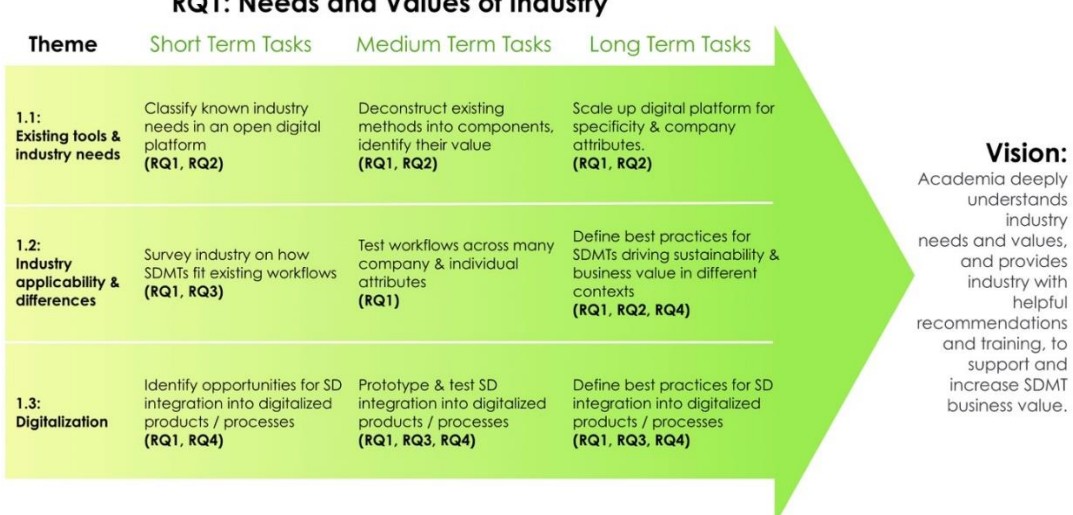

**Figure 5.** Research roadmap for RQ1: What are the needs and values of industry regarding sustainable design?

### 5.1.1. Theme 1.1: Existing Tools & Industry Needs

The short-term goal is to classify known industry needs for SDMTs, and what SDMTs are currently used successfully, in an openly accessible platform. This could be a centralized database or distributed forums, or collections of media, as long as it is easy for researchers and practitioners to access and contribute. It should include working with regulators, NGOs and, communities to understand their needs as well, since they often provide incentives and drivers for industry. Here, researchers should update literature reviews to collect, synthesize, group, and make findings available to help align the research community. This should include successfully-met needs as well as unmet needs. This includes what industry values and criticizes in existing SDMTs, especially in the context of:

- impact assessment and sustainability performance indicators
- goal setting and guiding solution development (e.g., Sustainability Design Space approach by Hallstedt [64])
- ideating
- comparing alternatives (deciding among tradeoffs)
- redefining problems
- communicating the value of sustainability (e.g., Life Cycle Costing)
- balancing environmental, social, and economic factors

The medium-term goal is to deconstruct existing methods into components and identify their value. This should involve breaking down SDMTs into component activities, mindsets, and tools, and then assessing:

- components that are most valued for goal-setting, ideation, and other business benefits (e.g., cost-saving, innovation, quality)
- components that are most valued for sustainability
- types of perceived sustainability value (e.g., energy efficiency, circularity)

- how perceived value compares to empirical assessments of using different design practices
- disadvantages or criticisms of components

All of these factors should be tested across a wide variety of industries, company types, and job roles, to determine if and how these circumstances influence the results.

The long-term goal is to scale up the digital platform from the short-term goal to include the specificity and industry data gathered in the medium-term goal. The result should make SDMT recommendations and learning widely accessible to and easily understandable by industry designers, engineers, and managers. It should also include concrete examples of cases and any important differences by company attributes.

### 5.1.2. Theme 1.2: Industry Applicability & Differences

The short-term goal is to survey multiple industries about how existing SDMTs currently fit into existing workflows, regulatory compliance, and organizational structures. As with the classification of SDMTs described above, this includes both successful implementations and unmet needs. It also includes what design team capabilities are needed on different organizational levels.

The medium-term goal is to empirically test SDMTs in workflows across a wide range of companies and decision-makers with different attributes to improve SDMTs' generalizability or specificity, as mentioned in Theme 1.1. This will help identify which SDMTs and industry needs are generalizable versus which require customization. These should examine differences in company types (e.g., manufacturer, consultancy), company sizes (large, medium, and small), industry sectors (e.g., electronics, apparel, furniture), job roles (e.g., designer, engineer, manager, sustainability specialist), individual demographics (e.g., gender, age, nationality), and geographic and/or political context.

The long-term goal is to investigate and define evidence-based, generalizable best practices for how SDMTs can drive sustainability and business value in different industry contexts. This uses results from short- and medium-term studies to show demonstration cases of where and how sustainability can reduce costs, increase margins, increase product quality, reduce risk, ease manufacturing and control of the supply chain, and increase innovation.

### 5.1.3. Theme 1.3: SDMT Digitalization

As industry moves to more digitalized product development practices, SDMT development must take into account these changes and the needs that evolve along the way. The short-term goal is to identify opportunities to integrate sustainability data and SDMTs into digitalized products and design processes. As digitalization becomes more ubiquitous, the amount of data, and data from many sources, need to be managed, sorted, and tracked for the product's life cycle [66]. Integrating data and analytics into design processes to facilitate product traceability and exchange data between stakeholders can drive system sustainability [18]. One example is the potential for integration of value and sustainability assessment in design space exploration by machine learning [67].

The medium-term goal is to clarify industry needs, opportunities, and barriers around sustainability in digitalized products and design processes, by prototyping and testing integrations of SDMTs. This includes developing concrete cases with industry relating to SDMTs in digitalized product development.

The long-term goal is to define evidence-based, generalizable best practices for integrating digitalized sustainability data from SDMTs into product life cycle management systems, to enable traceability of sustainability data and a cyclic usage of technical and natural resources. This includes tracking ecological and social responsibilities, both direct and indirect, for material extraction, manufacturing, transportation, usage, and end-of-life.

### 5.2. RQ2: What Improvements in SDMTs Would Most Drive Industry Forward?

RQ2 addresses the question of where industry needs are already known but unfulfilled, or where business needs are met but sustainability is weak. In this aspect of the vision, companies have a set

of SDMTs that product development teams use regularly, integrated into their standard product development process, and these practices provide both substantive sustainability improvement and other business benefits, such as innovation or cost-cutting.

Sustainable design methods, tools, or other practices should be developed to meet one or more of the following criteria: (i) easy to implement but rigorous whole-system view of sustainability, (ii) provide decision support, (iii) provide business value, (iv) help practitioners select and/or customize tools for their needs, and (v) support communication about sustainability between product developers and customers. All of these factors could be addressed in parallel, or in various orders. A timeline is suggested below to help set priorities. Three main themes are proposed for this RQ: (i) new SDMTs to meet the current and new demands; (ii) new approaches to integrate with business strategy and values; and (iii) new approaches to support communication about sustainability (See Figure 6).

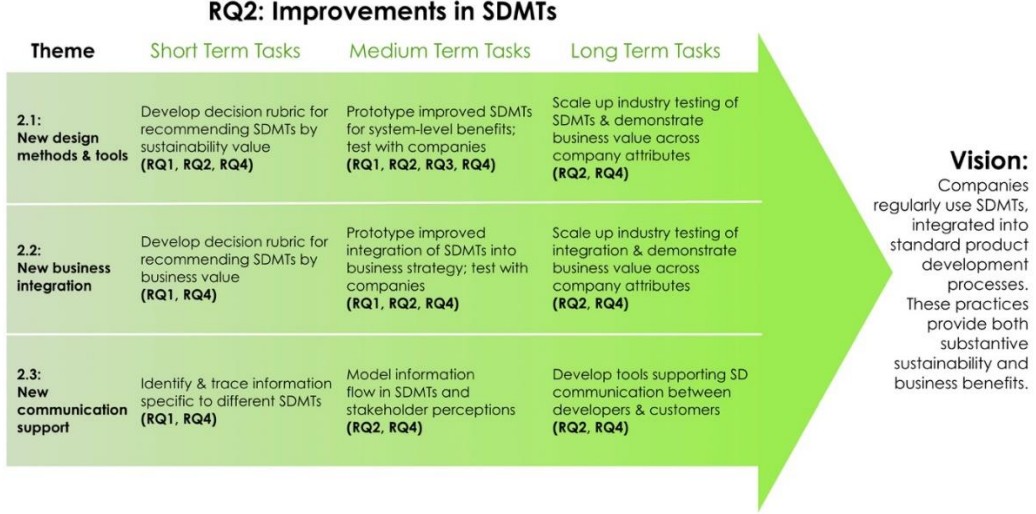

**Figure 6.** Research roadmap for **RQ2**: What improvements in SDMTs would most drive industry forward?

### 5.2.1. Theme 2.1: New Design Methods and Tools

The short-term goal is to develop a decision rubric for choosing SDMTs to use under given design scenarios, to optimize sustainability value. Tools of this kind exist in the form of lists or classification schemes, but they are scattered, contradictory, and not comprehensive. The research needed here is to harmonize existing classification schemes and gather comprehensive data on enough SDMTs to help practitioners choose the best design practice(s) for their circumstances. This could use data from Theme 1.1 on successful SDMTs and from Theme 1.2 on successful integration into workflow. This tool may recommend an SDMT, or components thereof, by characteristics such as product development stage (e.g., research, ideation, detailed design), short-term versus long-term perspective, qualitative versus quantitative requirements, and other differences. It should be affordable and widely available, ideally free and open access online. It should enable product development teams to discover and learn new design practices with minimal effort. Likely no such tool will ever capture all SDMTs, especially not those informally created or improvised by individual practitioners, but it should help practitioners find SDMTs proven to be effective and inspiring to industry. It should also encourage product development teams to communicate feedback on real-world testing, use, and implementation of the design practices, to support continuous improvement of the practices.

The medium-term goal is to prototype easy-to-implement but rigorous SDMTs, both environmental and social, that improve whole-system impacts, not improving one part to the detriment of the whole. Researchers and the research community should test and iterate on SDMTs

with multiple companies in different industries, for a balance of project manageability with generalizability. It may be possible for the community to define standard test cases used for comparing SDMTs, as is practiced in optimization research and computer science code development. While older SDMTs and improvised ones will still exist, these new SDMTs should raise the level of available tools. They will do so by meeting one or several of the following criteria:

- Be accessible to non-experts (list actionable conclusions, not merely display data)
- Be rigorous, data-driven, and transparent about sustainability impacts, enabling users to understand the reasons for the actionable "answers" by examining the data. This includes accompanying uncertainties, assumptions, and choices.
- Integrate well with existing workflows.
- Provide advice relevant to the job at hand, not just generic guidance.
- Be inexpensive or free, to maximize adoption.
- Be easily available and accessible, requiring minimal to no installation.
- Integrate environmental, social, and economic factors in concrete ways providing specific design recommendations.
- Enable comparisons among different design options, for decision-making.
- Enable comparisons of tradeoffs or synergies between different metrics (e.g., environmental harm versus social good, or identifying environmental benefits that also save money) at the level of the whole system. Tradeoff analyses should also enable modeling of rebound effects (e.g., cost savings from energy efficiency causing customers to use the product more, unintentionally causing increased energy use)
- Some design practices should combine forecasting (which is commonly seen) and backcasting (which can better help accomplish bold goals)
- Suggest not only design decisions, but also business model decisions, because many designs are economically non-viable without an enabling business model.

The long-term goal is to scale up the testing and iteration to dozens or hundreds of companies across the wide variety of company attributes listed in Theme 1.2. This will enable researchers to assess the generalizability of value (both for sustainability and business) in SDMTs or their components, and how to maximize value by combining or customizing SDMTs by industry, company, or other context. These results should be used to update and expand the decision rubric developed in the short-term goal.

5.2.2. Theme 2.2: New Business Integration

The short-term goal is to develop a decision rubric for choosing SDMTs by their potential business value in a company workflow. These rules would be most effective if they used data from Theme 1.2 on successful integration into workflow. They would also be most effective if they were incorporated into the tool described in Theme 2.1, including data and recommendations for business value, and if they included recommendations not just of standalone SDMTs, but where and when they best integrate into existing company product development practices.

The medium-term goal is to prototype integration of SDMTs into business strategy, relating low-level decisions to whole system impacts, balancing trade-offs, and suggesting business models to align profits with environmental and social benefits. Such integration is needed because many of the most important sustainable design decisions are made by business managers, before design teams are even assigned. Designers and engineers may be able to influence business strategy later, but must find ways to integrate such changes. Researchers should work with industry to prototype different integrations into business strategy, portfolio development, and product development workflows. These should meet one or several of the following criteria:

- Bring sustainable design options and thinking into pre-design business strategy decisions, so executives consider sustainability from the very beginning.
- Show design teams how detailed decisions change overall whole system impacts, and suggest design decision(s).

- Align low-level tactics / operations with the company's high-level global strategy.
- Provide business benefits with low sustainability risks, e.g., save money, increase profit margin, ease regulatory compliance, reduce lawsuit risk, improve marketing, enhance innovation.
- Demonstrate business value to design teams and executives, to drive adoption and integration.
- Communicate value to the customer, to drive market success.

Researchers should test and iterate prototypes of business integration with at least 3–5 companies in different industries, for a balance of affordable/manageable projects and generalizability. Prototypes with promising test results should be added to the tool that recommends design methods/tools via a decision rubric, to publicize them to interested practitioners.

The long-term goal is to scale up industry testing to refine integration into business strategy and demonstrate how sustainability can also provide business value, generalized across different types of companies. As with testing SDMT value, researchers should continue testing their integration into company business strategy, but scale up the testing and iteration to dozens or hundreds of companies across a wide variety of company characteristics. This data on the generalizability of SDMTs should be used to update and expand the decision rubric developed in the short-term goal, which recommends when and where to use each SDMT. Direct feedback on the decision rubric from practitioners can also be combined with other research to drive progress.

### 5.2.3. Theme 2.3: New Communication Support

The short-term goal is to identify and trace relevant sustainability information specific to different sustainable design processes, bridging the communication gap that exists between product designers and their customers [68]. Researchers here would identify the information customers need to understand a product's or service's sustainability benefits and tradeoffs, as well as the information product developers need to understand customer preferences. Researchers would also create or suggest information management tools to help companies track this information, to support strategic decision-making by designers and customers.

The medium-term goal is to model sustainability information flow in relation to sustainable design processes and various stakeholders' perceptions. Researchers would identify the sustainability needs and preferences of customers, supply chain vendors, marketing, management, legal department, and other stakeholders. Researchers would then create models of how these needs and preferences relate to one another and affect decisions in different stages of product design, development and consumption. They would also model how sustainability benefits, trade-offs, and suggested usage are communicated from design teams to customers to drive more sustainable purchasing and lifetime usage patterns, as well as communicated to supply chain vendors, marketing, management, and others, to ensure proper implementation of design intent and to harmonize with company business strategies.

The long-term goal is to develop SDMTs to support communication about sustainability between product developers and customers. SDMTs would be developed to support communication about sustainability between product developers and customers. They should meet one or more of the following criteria:

- Communicate sustainability information to customers and users, to demonstrate the value of the sustainability strategies or accomplishments embodied in the product [36,37,69].
- Act as decision support for customers and users, educating and motivating them to make more sustainable purchasing and usage decisions.
- Provide feedback from customers and users to design teams, enabling design teams to better meet their needs and values while continuously improving sustainability.

### 5.3. RQ3: How Should Researchers Move Forward With Developing Useful SDMTs?

RQ3 addresses the question of how researchers should interact with industry to find needs, prototype new SDMTs or consolidations of existing SDMTs, and user-test them for improvement. In

this aspect of the vision, researchers work closely with companies to iteratively co-create practices that enhance sustainability and business value.

RQ3 builds on RQ1 and RQ2. It both uses RQ1ä9s understanding of needs and values of industry and provides a means to deepen that understanding through a human-centered design approach [70] to design methods [71]. It provides a method for the development of SDMTs described in RQ2, and also treats RQ2's methods and tools as prototypes for user testing and iteration, proposing cycles of continuous improvement and co-creation with industry. This should enable the development of more relevant, usable, and useful tools. Four main themes are proposed for this RQ: (i) a human-centered design approach, co-creating with industry practitioners; (ii) researching and developing not only tools and methods themselves, but their integration into industry practice; (iii) how SDMTs succeed in the long term in industry practice; (iv) consolidating today's excess of SDMTs by identifying redundancies or synergies (See Figure 7).

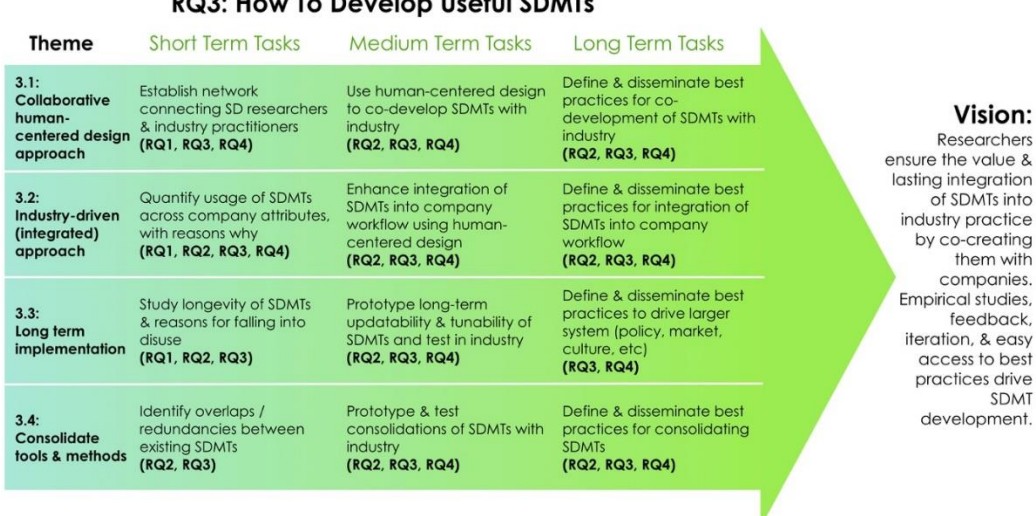

**Figure 7.** Research roadmap for **RQ3**: How should researchers move forward with developing useful SDMTs?

### 5.3.1. Theme 3.1: Collaborative Human-Centered Design Approach

The short-term goal is to establish a formal open network connecting SD researchers and industry practitioners. To test or co-create SDMTs using a human-centered design process, researchers must work with relevant industry participants; a network enables this. The network should be open and should include designers, engineers, and managers from a wide variety of industry sectors and company types, to enable generalizable solutions. Such a network will also provide industry practitioners with connections to academia, which may assist RQ4 through added training.

The medium-term goal is to iteratively co-create new, updated, and combined SDMTs with industry. Human-centered design would be used to target top-priority industry needs from RQ1, and treat existing SDMTs as prototypes to test, recombining the best aspects to form new or updated SDMTs in an iterative process, involving industry at every step. Also, remembering that practitioners generally do not practice SDMTs as taught, but use some components without others, research would investigate how mixing and matching components from different methods can add value. This could consolidate and streamline existing SDMTs, or customize them by context.

The long-term goal is to define and disseminate evidence-based, generalizable best practices for co-development of SDMTs with industry. The short-term and medium-term tasks would establish ample evidence of how the human-centered design process can improve SDMTs, and customize them by context. Once established, sharing best practices with researchers and industrial practitioners would help catalyze the dissemination of such practices into the mainstream of as many industries

as possible, also aiding RQ4. Definitions of best practices should be continually updated with additional research that accounts for changes in industry.

### 5.3.2. Theme 3.2: Industry Driven (Integrated) Approach

The short-term goal is to quantify actual usage rates of SDMTs in diverse companies, with rationales for that usage behavior, via global surveys. Such surveys are important to quantify what is working now and qualitatively find the reasons. Identifying failures, limitations, and barriers to existing SDMTs (e.g., [13]) should be included, but the focus should also be on best practices and examples of what is working well. This would be investigated across multiple company and individual attributes, including job role (e.g., designer, engineer, manager, sustainability specialist), company type (manufacturer or consultancy), company size, company longevity (startup versus established), product sector (e.g., consumer electronics, apparel, furniture), and even individual demographics like gender and age. Specific company design and production processes may also be important variables.

The medium-term goal is to test and enhance integration of SDMTs into company workflow using human-centered design. Researchers would use the same human-centered design approach described above to treat different integrations of SDMTs into company workflow as prototypes. Testing and iterating these prototypes can produce new and better integrations of SDMTs into workflows. It can also help customize them by industry, company, individual, or other circumstances. Such co-creation of SDMT integrations would not only help meet industry's needs better, but also build buy-in and help address unforeseen obstacles.

The long-term goal is to define and disseminate evidence-based, generalizable best practices for integrating SDMTs into industry product development practice. Similar to Task 3.1, this would build on the short-term and medium-term tasks that illuminate how the human-centered design process can improve SDMT integration in different contexts. When these best practices are shared with researchers and industrial practitioners, they will be disseminated into the mainstream of as many industries as possible, also aiding RQ4.

### 5.3.3. Theme 3.3: Long-Term Implementation

The short-term goal is to study longevity of existing SDMTs, and reasons why they fall out of practice. Researchers should not just consider short-term trials of SDMTs in industry, but further quantify the longevity of SDMTs, and qualitatively find the reasons why some slip out of practice, while others continue long-term. Such studies would also benefit from individual and company attribute specificity mentioned above.

The medium-term goal is to prototype a means to make SDMTs updatable to fit company needs in the long term and tunable to fit needs of different groups or individuals in a company, testing with industry case studies. Because development of SDMTs often stops after the exploration stage, most SDMTs stop being used after brief trial periods. Therefore, in these cases, much of the work developing SDMTs is wasted. Researchers can multiply their impact by leveraging existing foundations to continue refining, developing, and implementing SDMTs. This includes both research into updating and tuning existing SDMTs and research into how to make SDMTs inherently updatable and tunable without external support by researchers.

The long-term goal is to define and disseminate best practices for creating SDMTs to be updatable, tunable, and generally sustainable for long-term implementation. The short-term and medium-term tasks would provide data to establish what provides the most leverage in updating and tuning SDMTs. These can be used to develop a framework for how to make SDMTs updatable and tunable. This framework of best practices would then be disseminated to the research community and industry, both to drive better SDMTs and to drive the process of developing SDMTs from the "one-shot case study" to more long-term sustainable implementation.

### 5.3.4. Theme 3.4: Consolidate Tools and Methods

The short-term goal is to identify overlaps and redundancies between existing SDMTs to avoid continually reinventing the wheel, as has so often been done in SDMTs. This may include categorizing and sorting SDMTs, connecting with RQ1's short-term goal to classify known industry needs in an open digital platform (centralized database, media forums, or other), and RQ2's short-term goal to develop a decision rubric for when and where to use specific SDMTs. Then, all new SDMTs could be framed with their commonalities to other existing SDMTs as well as their unique contributions. This would make it easier for companies to adopt, modify, and combine SDMTs, and help integrate them into existing workflows.

The medium-term goal is to prototype and test consolidations of existing SDMTs to streamline them, using human-centered design approaches with industry. As mentioned above, this approach treats existing SDMTs as prototypes for user testing and iteration; in product design, this process drives consolidation of many initial prototypes into one best product, or a few best products for different companies or circumstances. This process can also drive consolidation of today's vast array of SDMTs into a smaller set of more effective tools. This research should combine with RQ1 and RQ2's open-access platforms to list and recommend existing SDMTs for different industry needs. The prototyping and testing should employ empirical case studies testing a broad range of company attributes and should include challenging test cases that exemplify obstacles to adoption.

The long-term goal is to use the short- and medium-term results to define and disseminate best practices for consolidating some SDMTs and framing others in the context of the suite of available SDMTs. These best practices should include accommodation to different industries, companies, individuals, or other circumstances. This framework of best practices should, as with this topic's previous tasks, use RQ1 and RQ2's open platform to be disseminated to researchers and industry. The goal is to avoid over-proliferation of new but ineffectual SDMTs and instead drive fewer SDMTs that are widely adopted and deeply integrated into standard industry product design to create large-scale tangible sustainability benefits.

### 5.4. RQ4: How can Sustainable Design be More Effectively Integrated Into Industry?

The vision of this article includes a future in which companies integrate sustainability thinking deeply throughout their entire product and service development processes [72], helped by the latest academic research in the field, to achieve measurable lasting change at a global scale. A timeline is presented in Figure 8 to show the five themes and corresponding research guidance for achieving this vision. This requires a clearer understanding of the (i) implementation of SDMTs in organizations and their product development practice, (ii) alignment of SDMTs with business incentives and barriers, and (iii) interactions with external systemic factors. Additionally, researchers need to be able to effectively (iv) train industry practitioners and disseminate knowledge in the latest SDMTs, and (v) open doors for education-industry collaboration, to facilitate impactful and lasting change.

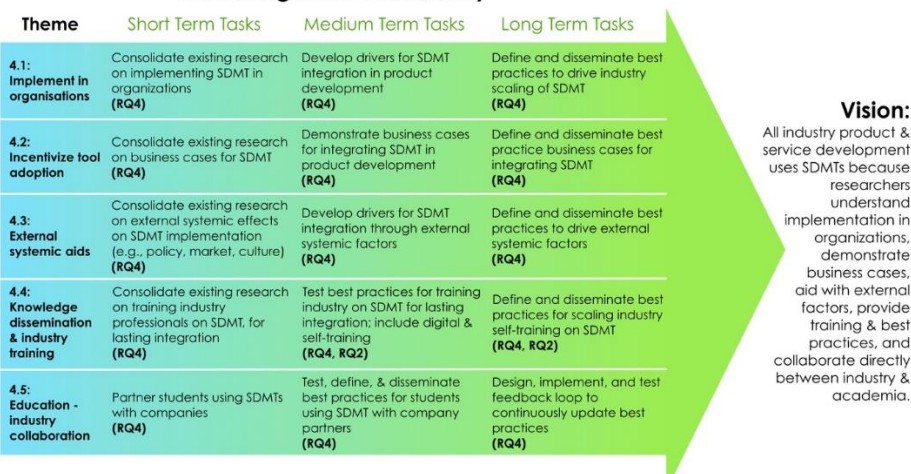

**Figure 8.** Research roadmap for **RQ4**: How can sustainable design be more effectively integrated into industry?

### 5.4.1. Theme 4.1: Implement in Organizations

The short-term goal is to collect and consolidate existing research findings on SDMT implementation in organizations and companies. Continued investigations are needed to deepen our understanding of sustainability implementation in relation to risk assessment, portfolio management, and product requirements. Specific topics include:

- What are the primary barriers to industry integrating SDMTs? What are the primary drivers?
- What are the main differences in integrating SDMTs across company attributes and circumstances? (E.g., small vs. large organizations; manufacturer vs. consultancy; industry sector; geographic, cultural, or political context; and sustainability maturity.)
- What constraints must be worked within, and what constraints can be overcome?

The medium-term goal is to build on this knowledge to develop and test theoretical models and operational frameworks for integrating SDMTs in business goals and workflow, despite organizational barriers. Psychology research has shown that humans usually consider risk and reward not by statistics but by their ability to remember relevant instances [73]. Therefore, publications of success stories can play a powerful role in driving future SDMT adoption, which would require more studies of successful integrations. This would help overcome the bias toward status quo product development practices [74].

The long-term goal is to establish evidence-based, generalizable best practices for SDMT implementation. This includes successful cases that connect to stakeholder values and solve real world problems effectively. These would apply various academically grounded lenses, such as business models, risk management, portfolio management, and product requirements.

### 5.4.2. Theme 4.2: Incentivize Tool Adoption

The short-term goal is to encourage companies or organizations to adopt SDMTs by consolidating existing research findings on business cases, such as:

- How much does trustworthiness play a role in decisions to use a tool (or not)?
- What can we learn from existing successful business cases of SDMT implementation?
- How can we modify business accountability methods to include a full vision of the value created by a business, rather than purely focusing on financial profits?
- How can companies assess the rebound effects of their design decisions?

The medium-term goal is to demonstrate successful business cases that adopt SDMTs. Theoretical models, organizational change strategies, and operational frameworks on these topics can be developed for integrating SDMTs into business workflow, incentives, and disincentives.

The long-term goal is to establish evidence-based, generalizable best practices for incentivizing SDMT adoption from within companies. This includes disseminating business cases that balance short-term profits and long-term sustainability goals for scaling up the integration of SDMTs. These best practices should include accommodation to different company attributes or other circumstances.

### 5.4.3. Theme 4.3: Aid with External Systemic Factors

The short-term goal is to consolidate existing research on how external systemic factors such as government regulations affect SDMT implementation. For example, this may include investigating how SDMT integration is driven or hampered by national or local government policies, NGOs, local communities, industry certifications, market trends in demand, cultural values, and other factors.

The medium-term goal is to use the consolidated knowledge of external systemic factors to develop programs that drive adoption and implementation of SDMTs in companies. This may include prototyping and testing of legislative policies, certifications, marketing campaigns, lawsuits, or other external factors.

The long-term goal is to establish evidence-based, generalizable best practices for incentivizing SDMT adoption in companies through external systemic factors. This may include case studies of successes in markets, government policies, lawsuits, social and cultural changes, or may include actual legislation, marketing campaigns, and other initiatives.

### 5.4.4. Theme 4.4: Industry Training

The short-term goal is to review the state of the art in industry training of sustainable design, to support more effective training that leads to higher adoption and more lasting implementation of SDMTs. This will provide an evidence-based foundation to build upon for training industry practitioners on SDMTs.

The medium-term goal is to test and enhance best practices for industrial training on SDMTs for lasting integration. Industry training best practices would be systematically tested for their effectiveness in researcher-led training sessions as well as self-training modules. This step would gather further evidence on the effectiveness of the leading approaches to training from the literature, by studying participant attitudes as well as longitudinally tracking changes in sustainable design practices. This would test both in-person training methods and self-guided online training methods. SDMT education would be provided not only for future designers and engineers, but also management professionals. This would be facilitated through workshops and other communication channels, with the goal to make people aware that sustainable design is more than eco-design but also social sustainability, management, and decision-making, and that SDMTs can be applied not only in the product design stage but on a strategic level. Digital and information technology tools are potentially effective means to ease SDMT knowledge dissemination and to support industry self-training.

The long-term goal is to use the results of the short- and medium-term tasks to develop best practices for industry practitioners teaching SDMTs to themselves and one another, in order to scale SDMT training to reach all industries globally. To truly integrate SDMTs into product development, we cannot be limited by the number of academics teaching; rather, we need professionals to also train themselves and one another. Online training tools should be updated and posted to a centralized and publicized repository, and teacher guides for in-person training should be disseminated widely.

### 5.4.5. Theme 4.5: Education-Industry Collaboration

The short-term goal is for academic researchers to influence practitioners towards the use of SDMTs through academia-industry collaborations. Academics frequently perform class projects or case studies that apply SDMTs to real industry problems, which not only trains students but also

provides direct sustainability recommendations to industry. In addition, as the students learn SDMTs and apply them to the product, their communication with company liaisons teaches SDMTs to the industry professionals to some extent. Currently, there are no best practices for how to perform such projects or studies, so existing research should be reviewed in the near term to compile generalizable findings.

The medium-term goal is to test the best practices for academia-industry collaboration, to find which approaches lead to measurable sustainability improvement in company products, stronger cooperation and collaboration, industry influence, and training of professionals in SDMTs through student partnerships. For example, open-ended student research projects may give more influence than structured class projects in certain circumstances, but vice-versa in other circumstances.

The long-term goal is to continuously update evidence-based best practices through the iterative process of designing education-industry collaboration, testing, and enhancing practices.

## 6. Discussion

The long list of research questions (RQs), themes, and tasks above may seem overly ambitious for the time desired, but any one of the roadmap tracks could be easily achieved with a motivated academic team and company partners. Whether all of the roadmap items are achieved in time simply depends on how many academics and companies are active, and how well they collaborate. For example, if one graduate student focused on each short-term task, all tasks could be achieved to an adequately useful degree in two to four years with 15 students, some with several industry partners. Indeed, fewer researchers are likely needed, because of the many overlaps among tasks. Performing these tasks and building the necessary industry connections would prepare researchers to scale up to medium and long term tasks, and evaluate their probability of success. Progress on tasks can be accelerated and fostered through cross-institute collaborations and idea sharing within the sustainability and design research communities.

Because all the research tasks are intertwined, achieving a task in one theme of one research question usually helps achieve tasks in other themes of the other research questions. To help the reader navigate through the roadmap and understand how different RQs and their subtopics are intertwined, a dependency structure matrix (DSM) is provided in Figure 9 to link the 15 themes of the roadmap. It was generated by one author, then all other authors checked it independently, then the group used one videoconference to collectively edit and finalize it. The DSM enables a clear and concise representation of complex systems or development processes [75]. The MICMAC (matrix impact cross-reference multiplication applied to a classification) method [76] was used to show the degree of connection between the themes and tasks of the roadmap as follows: one dot "•" for a potential connection, two dots "••" for an indirect/implicit connection, and three dots "•••" for a direct/explicit connection. For instance, the key themes 2.2 "new business integration" and 3.2 "industry-driven approach" are directly intercorrelated (strong mutual connection) because 3.3's human-centered design process will naturally lead to 2.2's outcomes of having tested new or modified SDMT prototypes with industry and assessed improvements, and achieving 2.2's outcomes will require at least a roughly human-centered design approach. Accomplishing tasks in the themes 2.2 and 3.2 could also be catalyzed to different degrees by achieving the tasks respectively from themes 1.1 "existing tools and industry needs", 1.2 "industry applicability and differences", and 1.3 "digitalization of SDMTs". In turn, accomplishing 2.2 and 3.2 would catalyze theme 4.1 "implement in organizations" by both improving the value of SDMTs and by the process of working with companies to improve SDMTs. Such interconnections and positive feedback loops will help build momentum for sustainable design researchers following the roadmap. Such interconnections also mean that coordination among researchers will help drive progress substantially.

In general, the short-term tasks emphasize collecting and consolidating information and previous work consistently and transparently. Many of these tasks address multiple research questions (RQs), because advancing SD requires a cross-cutting approach; however, they were presented in this paper by their primary RQ association. To clarify the needs and values of industry (RQ1), the short-term tasks involve classifying known information, surveying practitioners, and

identifying digitalization opportunities for SDMTs. To advance development of SDMTs (RQ2), the short-term tasks include identifying sustainability information that would be important to communicate between designers and customers, and developing decision rubrics to help practitioners identify the most appropriate SDMTs for their business and sustainability needs. To advance the SDMT development process (RQ3), short-term tasks establish an open network among diverse SD professionals, quantify usage rates and longevity of SDMTs, and identify redundancies in existing practices. Finally, to promote industry integration (RQ4), the immediate tasks are to consolidate information on implementation practices, case studies, and training.

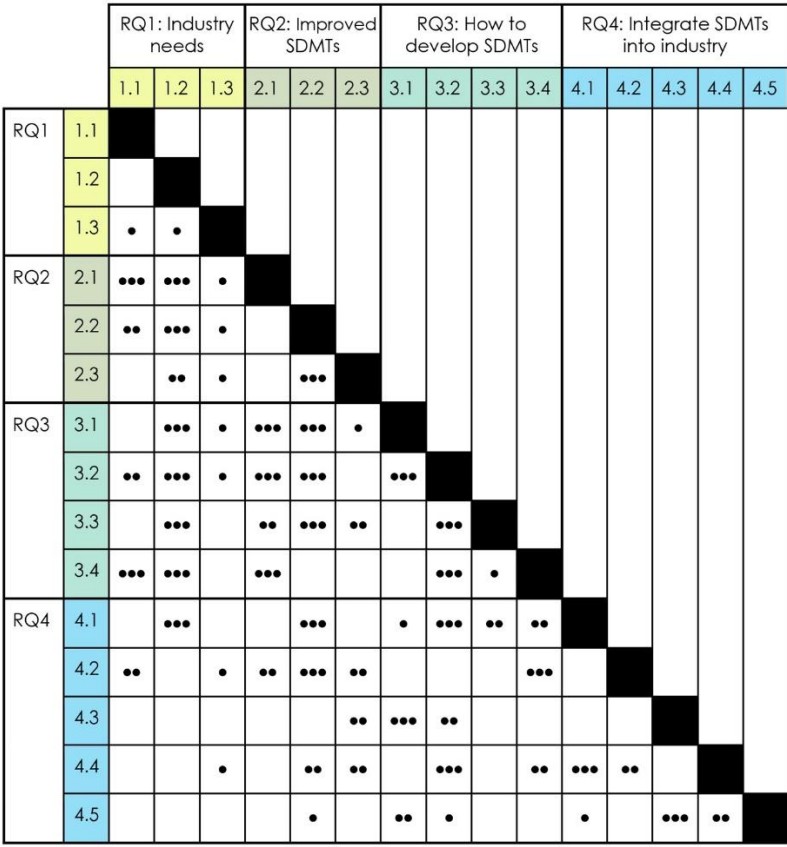

**Figure 9.** Dependency structure matrix (DSM) of the themes and tasks of the SDMTs roadmap, where one dot indicates a potential connection, two dots indicate an indirect/implicit connection, and three dots indicate a direct/explicit connection.

These short-term tasks will help the research community ramp up to larger and more influential medium-term tasks, improving SDMTs and their adoption with industry stakeholders. To better understand the values of industry (RQ1), these medium-term tasks will deconstruct and analyze SDMTs to identify core components of value, and they will test the integration of SDMTs and digitalization approaches within existing industry workflows. To drive SDMT development for practitioners (RQ2) and the processes by which researchers develop them (RQ3), the medium-term tasks include prototyping and testing SDMT innovations, consolidations, and integrations into company practices; modeling information flows in SD processes; and using human-centered design for co-development of these SDMTs. To advance industry integration (RQ4), the medium-term tasks will identify and develop drivers for SDMTs, and they will demonstrate and test best practices for SDMT dissemination and integration.

Finally, the long-term tasks that will lead to achieving the vision of ubiquitous sustainable design primarily involve scaling up industry SDMT deployment and testing, as well as defining, testing, and disseminating best practices. To solidify a common understanding of the needs and values of industry (RQ1), the long-term tasks will create a full-scale digital platform that contains

existing SDMTs and needs, sortable by specific characteristics. Additionally, best practices will be refined and put forward to enable SDMTs to drive business value and integrate into increasingly digitalized design and manufacturing environments. To promote stronger SDMT development (RQ2), we suggest scaled-up industry testing of improved SDMTs to demonstrate and document business value across company attributes and circumstances. To better enable sustainable design that meets customer needs, communication tools should be developed and deployed that enable bidirectional feedback between product developers and their customers. To improve the means of developing SDMTs (RQ3), long-term tasks will define and disseminate best practices to enable co-development, integration with company workflows, long-term implementation, and improvement and consolidation of SDMTs. To deeply integrate SDMTs into industry (RQ4), the long-term tasks will define and disseminate best practices to drive industry integration and scaling, handle variability in external factors, and infuse SD into education. To ensure that these results are sustainable in the long-term, a feedback system is recommended to continuously update these best practices with new research findings.

The scope of this research roadmap is limited to focus on the academic study of developing and integrating improved SDMTs, which are intended for use by industry practitioners. There are certainly other sources, users, and drivers of SDMTs, such as governments, nonprofits, and community organizations. The current findings can also be used by NGO and government practitioners, and we hope that other researchers will continue to explore unique aspects and opportunities of NGOs and government in the development and application of SDMTs. Furthermore, there are economic and cultural factors that significantly contribute to SDMT adoption and success, which are also outside the scope of this work. A roadmap for other stakeholders and disciplines, and their relationships with sustainability-focused decision support tools, would be a useful future exercise, and the authors recommend that such efforts be undertaken in collaboration with relevant non-academic and non-designer stakeholders.

Nevertheless, the scope is quite ambitious for the limited time before 2030, and the academic community will face challenges in achieving these goals and visions. Much of the success of achieving this roadmap relies on ample and effective communication—both among the research community and between researchers and practitioners. Organizations like the Design Society, ASME, and others can play a key role in providing or facilitating communication hubs, events, and publications. This can help share knowledge and SDMTs, reduce redundant research, and enable different research groups to more efficiently build on the recent work of others. While this research roadmap has the potential to raise the degree to which SDMTs drive substantive sustainability, business viability, and creative inspiration, we expect that informal, improvised SDMTs will continue to exist, and not every SDMT will meet all of the criteria laid out in this vision.

## 7. Conclusions

Industry must reorient itself for sustainability, and design is a key enabler to do so. While hundreds of sustainable design methods and tools (SDMTs) have been made available by the design research community, industry has not yet embraced sustainable design as common practice. There are several well-documented reasons for the low adoption of SDMTs, including the decentralized nature of research, its disconnection from industry, and the misalignment of SD with business strategies. A new strategy for more effective research and development is required.

The research roadmap presented in this paper was generated through a multi-year process with an international team of academic and industrial researchers that included a review of existing practices and theory, elicitation of expert knowledge, structured brainstorming, and iterative development of an overarching vision and specific tasks. It offers a new strategy for more effective SDMT research, development, and implementation. Its pathway will resolve four high-level research questions through 15 themes, each with multiple tasks, which contribute to achieving a future vision in which sustainable design is ubiquitously and consistently practiced in industry. In the context of academic training, each of these themes alone might inspire an entire or partial PhD dissertation, or individual tasks may be master's thesis projects. Given the scope and breadth of the suggested work,

it will take the efforts and expertise of many research teams to accomplish these goals. The tasks are organized to include immediately actionable short-term tasks, built upon by medium-term tasks, and finally crowned by long-term tasks to attain the ideal vision described in Section 3.

This research roadmap is intended to enable more tangible success in mainstreaming sustainable design practices in industry, ensure that such practices provide more sustainability and business value, and facilitate academia in better understanding the needs of design tool users and creators. It should also help academics decide what projects or research questions to pursue, both for developing better methods or tools and for recommending existing ones more effectively. While the overarching goal is to transform industry, the roadmap is aimed at the design research community. It assists in prioritizing future research tasks, strategically planning research projects, and strategically planning funding applications. The roadmap can aid in identifying discussion and interest groups for international collaborations, enabling less redundancy and more efficiency across the research enterprise. The roadmap can also be used in discussions with industry and policymakers, to help structure and unify views on the necessary steps to convert industry production and consumption to sustainability. A concrete roadmap such as this can help the design community turn visions of sustainability into reality.

**Author Contributions:** Conceptualization, literature review, analysis, roadmap development, and validation by all authors; writing and editing primarily by S.Y.K., S.H., S.I.H., M.S., and J.F.; final editing by J.F.; visualization by J.F., S.H., and M.S.; project administration by C.T. and S.H. All authors have read and agreed to the published version of the manuscript.

**Funding:** A portion of this study was funded by the Knowledge Foundation in Sweden.

**Acknowledgments:** Special thanks to the participants in the ICED 2017 workshop discussed, and to the expert reviewers of the roadmap. Thanks also to the Design Society for its support of the special interest group.

**Conflicts of Interest:** The authors declare no conflict of interest.

## Appendix A

Discussion questions for experts at DESIGN 2018 conference
For industry practitioners:

1. In your practice, what design methods, activities, or mindsets do you get the most value from?
2. Why do you (not) get value from these methods, activities, or mindsets?
3. In practitioners opinions, which of the design method's activities or mindsets improved product sustainability?
4. In practitioners opinions, did anything in the design method provide any other value, not related to innovation or sustainability?
5. What are practitioners expecting of eco-design methods and tools in the next few years? (e.g., computer-based, open source, more integrated, etc.)
6. In practitioner's experience, what effect does sustainability usually have on design? (Checklist: increases/decreases legal risk, increases/decreases design process cost, increases/decreases final product cost, restricts/enhances creativity, decreases/increases your motivation, complicates/eases manufacturing, decreases/increases product quality, decreases/increases product marketability)
7. What prevents practitioners from using sustainable tools?
8. What makes practitioners use/implement tools more?

For design method creators/researchers:

1. Are you currently working on the experimentation (of existing) or development (of new) eco-design tools?
2. If so, according to you, what are missing to meet your specific needs? (please be precise about these needs, e.g., industrial application, learning purposes)
3. Would it be of interest to have a road map to guide researchers of which areas to focus on in order to develop/improve sustainable design support tools in future?

4. Have you any ideas of what are some gaps today in relation to sustainable design tools?
5. What do professionals value in existing green design methods?
6. Specifically, where do they find sustainability value, innovation value, and other business value? (cost, legal risk, other design process, etc.)
7. What do they want that existing methods don't provide?
8. What outcomes result from different methods?
9. How can design methods align business incentives with environmental & social impacts?

## Appendix B

Primary research questions leading into ABCD methodology

1. In industrial practice, what design methods, activities, or mindsets do businesses/designers get the most value from? Why?
2. What do industrial practitioners/businesses want that doesn't exist?
3. How does industry receive education/training and start using tools?
4. What knowledge is missing to meet your specific needs in tool development and experimentation?
5. What do you think needs to change about the way we create tools and methods?
6. What are some gaps today in relation to sustainable design tools?

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
