# Peer review of "A Research Roadmap for Sustainable Design Methods and Tools"

_sustainability, doi:10.3390/su12198174_

Round 1

Reviewer 1 Report

Thank you for getting the opportunity to read and review a paper titled "A Research Roadmap For Sustainable Design Methods And Tools”.

I found this paper very interesting and I am very impressed with its content. The findings are of general interest and the paper fits the aims and scope of the “Sustainability” Journal.

This paper merits publication, although I believe it suffers from few weaknesses. Therefore, I suggest some recommendations for the Authors’ consideration in order to improve the reviewed manuscript.

  1. The content of the paper is impressive. However, it is not a policy document but scientific paper. Therefore, the description of its methodology should be a key part of it. Unfortunately, it is rather unclear for me. It is said, inter alia, that:
    • “This research was conducted by an international working group consisting of 8 academic and industrial researchers based on regular online meetings.” Please specify any details about these meetings, e.g. their numbers, duration etc.
    • “Working group reviewed the literature individually and then the findings were discussed as a group.” Please specify how it was organized, recorded, ordered etc.
    • “The review focused on seminal works in the field.” Please provide some examples of such seminal works.
    • “Additional articles were found through searches in Web of Science, Science Direct, and Google Scholar, as well as papers that either cited or were cited by those that were already part of the review.” Please specify how many, approximately, articles were reviewed, how they were categorized?
    • “Results were supplemented by informal interviews with other sustainable design researchers.” Please provide some information about the number of these interviews, the number of interviewed researchers, how these interviews were organized, recorded, analyzed?

  1. According to the Authors, “using previous interview experiences conducted by the authors, individual expertise, and the background literature, the working group generated 17 key questions. Could you describe any details how these questions were generated, who participated in this process, how many questions were proposed initially, etc.?

  1. Roadmap was defined using a backcasting approach. Please specify and provide some details how and by whom it was developed, step by step, on the timeline.

  1. Considering “Dependency structure matrix (DSM) of the themes and tasks of the SDMTs roadmap”, please specify how this matrix was constructed, e.g. how many experts participated in the process, how different opinions were agreed, etc.

  1. I know that it is an academic paper. However, we must be realistic. Therefore, in my opinion, utopian and unrealistic assumptions should be avoided. Do the Authors really believe that by 2030:
    • “All companies have a long-term sustainability strategy that is deeply integrated into their product development and other business practices so that their products and services enhance the environmental and social health of the world more than they damage it.”
    • “Companies regularly use SDMTs integrated into standard product development processes”
    • “All industry product & service development uses SDMTs…”

Unfortunately, I am afraid that it is mainly wishful thinking. Somehow, this reminds me of the Agenda 21 ambitious assumptions from 1992. We know how effective this action plan was executed although it was almost 30 years ago.

  1. The Authors proposed very ambitious short-, medium- and long-term tasks related to the four main research questions. It would be very interesting to discuss within the working group and with other sustainable design researchers how they assess the probability of effective implementation of each of the proposed tasks by 2030. This would also show the feasibility of the proposed roadmap.

Author Response

Please find the replies to both reviewers in the attached file.

Reviewer 2 Report

Please find the points below for improvement:

  • The paper is too long and more suitable as a book chapter at this stage.  Please shorten it to fit as a journal paper.
  • A case study should be included in the paper.
  • Add a flow chart to describe a methodology.

Author Response

(The authors gave the same response as above.)

Round 2

Reviewer 1 Report

I am not fully satisfied with the responses and introduced modifications, but I think the article is worth publishing.

Ultimately, the article’s readers will be its final and main reviewers.

However, I have the last question. The authors mentioned that the working group consisted of 8 academic and industrial researchers. Are all these experts among the authors of this article?